# Versatile and flexible microfluidic qPCR test for high-throughput SARS-CoV-2 and cellular response detection in nasopharyngeal swab samples

Julien Fassy[1☉], Caroline Lacoux[1☉], Sylvie Leroy[1,2], Latifa Noussair[3], Sylvain Hubac[4], Aurélien Degoutte[2], Georges Vassaux[5], Vianney Leclercq[6], David Rouquié[7], Charles-Hugo Marquette[2], Martin Rottman[3,8], Patrick Touron[4], Antoinette Lemoine[3], Jean-Louis Herrmann[3,8], Pascal Barbry[1], Jean-Louis Nahon[1], Laure-Emmanuelle Zaragosi[5], Bernard Mari[1]*

1 Université Côte d'Azur, CNRS, Institut de Pharmacologie Moléculaire et Cellulaire, FHU-OncoAge, Valbonne, France, 2 Département de Pneumologie, CHU-Nice, FHU-OncoAge, Université Côte d'Azur, Nice, France, 3 Assistance Publique-Hôpitaux de Paris, GHU Paris–Saclay, Garches, France, 4 Institut de Recherche Criminelle de la Gendarmerie Nationale (IRCGN), Cergy, France, 5 Université Côte d'Azur, INSERM, CNRS, Institut de Pharmacologie Moléculaire et Cellulaire, Valbonne, France, 6 LBM BIOESTEREL, Mouans-Sartoux, France, 7 Bayer SAS, Valbonne, France, 8 Université Paris-Saclay, UVSQ, Inserm, Infection et inflammation, Montigny-Le-Bretonneux, France

☉ These authors contributed equally to this work.
* mari@unice.fr

**Data Availability Statement:** All data are provided in supplemental information.

## Abstract

The emergence and quick spread of SARS-CoV-2 has pointed at a low capacity response for testing large populations in many countries, in line of material, technical and staff limitations. The traditional RT-qPCR diagnostic test remains the reference method and is by far the most widely used test. These assays are limited to a few probe sets, require large sample PCR reaction volumes, along with an expensive and time-consuming RNA extraction step. Here we describe a quantitative nanofluidic assay that overcomes some of these shortcomings, based on the Biomark™ instrument from Fluidigm. This system offers the possibility of performing 4608 qPCR end-points in a single run, equivalent to 192 clinical samples combined with 12 pairs of primers/probe sets in duplicate, thus allowing the monitoring of SARS-CoV-2 including the detection of specific SARS-CoV-2 variants, as well as the detection other pathogens and/or host cellular responses (virus receptors, response markers, microRNAs). The 10 nL-range volume of Biomark™ reactions is compatible with sensitive and reproducible reactions that can be easily and cost-effectively adapted to various RT-qPCR configurations and sets of primers/probe. Finally, we also evaluated the use of inactivating lysis buffers composed of various detergents in the presence or absence of proteinase K to assess the compatibility of these buffers with a direct reverse transcription enzymatic step and we propose several protocols, bypassing the need for RNA purification. We advocate that the combined utilization of an optimized processing buffer and a high-throughput real-time PCR device would contribute to improve the turn-around-time to deliver the test results to patients and increase the SARS-CoV-2 testing capacities.

**Funding:** Supported by funds from the "Centre National de la Recherche Scientifique" (CNRS), the "Université Côte d'Azur", the French "French Defence Innovation Agency – Agence de l'Innovation de Défense ("project "Safe and direct COV-2 qPCR Test") and the Département des Alpes Maritimes (COVID-19 Health program). JF is supported by the Cancéropole PACA and CL is supported by Plan Cancer 2018 « ARN non-codants en cancérologie: du fondamental au translationnel » (number 18CN045). P.B. is recipient of a chair from ANR-19-P3IA-0002–3IA Côte d'Azur–Nice–Interdisciplinary Institute for Artificial Intelligence. The Biomark equipment was funded by Canceropole PACA and France Génomique (Commissariat aux Grands Investissements: ANR-10-INBS-6 09–03, ANR-10-INBS-09–02). The funders had no role in study design, data collection and analysis, decision to publish, or preparation of the manuscript. Bayer and LBM Bioesterel provided support in the form of salaries for authors [DR and VL], but did not have any additional role in the study design, data collection and analysis, decision to publish, or preparation of the manuscript. The specific roles of these authors are articulated in the 'author contributions' section. Bayer SAS also loaned equipment and reagents for the study.

**Competing interests:** The authors have read the journal's policy and have the following competing interests: DR is a paid employee of Bayer SAS and VL is employed by LBM Bioesterel (LBM Bioesterel). There are no patents, products in development or marketed products associated with this research to declare. This does not alter our adherence to PLOS ONE policies on sharing data and materials.

## Introduction

To control the pandemic and monitor virus propagation of SARS-CoV-2 in real time, extensive testing is necessary. Although alternatives are available [1, 2], viral load detection from nasopharyngeal or saliva samples is still the most appropriate method to identify SARS-CoV-2 carriers. The current diagnostic testing methods recommended by the Centers for Diseases Control (CDC) and the World Health Organization (WHO) are based on a traditional RT-qPCR assay, with validated primers [3]. However, the availability of this assay has been a major hurdle in the orderly and efficient management of the pandemic because of reagent shortages, as well as material and staff limitations. A rapid processing of the samples is also a crucial factor. To further stem the spread of coronavirus disease 2019 (COVID-19), a series of innovative approaches have been recently proposed [4]. Notably, to face material shortage and reduce processing times, two kinds of process optimization should be undertaken: (1) multiplexing sample and PCR probes, as well as (2) reducing the steps in sample preparation.

Most approved RT-qPCR assays are "one-step" kits that should be performed on standard real-time thermocyclers. When sample pooling [5] is not used, throughput is quite limited and the use of alternative systems, such as the Biomark$^{TM}$ HD device from Fluidigm may overcome this limitation. The Biomark$^{TM}$ HD device is a nanofluidic automated real-time PCR system that exploits the microfluidic technology through the use of dynamic arrays of integrated fluidic circuits (IFCs). Using, for example, the 192.24 Integrated Fluid Circuit (IFC), 192 samples can be processed in parallel with 24 independent sets of probes, allowing increased confidence in viral RNA detection as additional SARS-CoV-2 probes can be included. Probes to detect supplementary targets such as other RNA viruses, or host response genes can also be included. This flexibility presents a clear added value for both clinical monitoring of viral and bacterial pathogens as well as for research studies [6, 7]. Another advantage of this system resides in the low amount of reagents needed compared to classical real-time PCR machines. The reaction volume is down to the 10 nL-scale compared to the 10 μL-scale in classical qPCR, thus reducing reagent requirement, an important point in times of shortages.

The second improvement resides in the reduction of the number of steps for sample preparation. RT-qPCR detection starts with viral RNA extraction. This is time consuming for laboratories that are not equipped with high-throughput automated systems. It is also reagent consuming, and as March 2020, laboratories have suffered from major RNA extraction kit shortages. Several teams have proposed RNA extraction-free protocols. The resulting improvement in processing times was balanced by a loss in sensitivity. The comparison of these different studies is made difficult by the great variabilities of the protocols. This includes parameters such as the type of collection media, the use of additives such as detergents, heat-inactivation, or selection of a particular RT-qPCR kit [8–16]. In 2 of these studies, measurements on a large set of clinical samples demonstrated the effectiveness of direct RT-PCR assays with or without heat inactivation on various transport medium (VTM) [8, 9]. Srivatsan et al. proposed to collect dry swabs and elute them in Tris-EDTA (TE) to eliminate collection medium dilution and variation, and to bypass the RNA extraction step [17]. This simplified protocol was as sensitive as a conventional protocol (sample collection in VTM). They also evaluated the addition of detergents (IGEPAL, Triton X-100 and Tween-20) that facilitates virus inactivation in the elution medium. Addition of all three detergents resulted in a loss of sensitivity.

In the present study, we evaluated the use of a high-throughput real-time PCR device, the Biomark$^{TM}$ HD, to increase throughput, flexibility in probe inclusion, and decrease reagent consumption, together with an optimized protocol for SARS-CoV-2 RNA detection without RNA extraction.

## Materials and methods

### Primers and probes for the detection of viral and cellular genes

All the DNA primers and probes used in this study are listed and described in Tables 1 and 2. The set of primers/probe used to detect the wild-type spike (S) gene was described in [18].

### Positive control plasmid

Positive control plasmid containing the full length nucleocapsid coding sequence (N) was purchased at Integrated DNA Technologies (2019-nCoV_N_Positive Control; cat. no. 10006625). Plasmid was supplied at a final concentration of 200 000 copies/μL in IDTE pH = 8 buffer, and has been used to evaluate both the detection efficiency and sensitivity of the Biomark$^{HD}$ System Analysis.

### *In vitro* transcription

The synthetic transcript for the 2019-nCoV N coding sequence (N transcript) was generated using the 2019-nCoV_N_Positive Control plasmid. The template for the *in vitro* transcription was produced by PCR: briefly, 6 ng of plasmid was used to amplify the entire coding sequence of the N gene using: T7-For-sens (ATATAATACGACTCACTATAGGATGTCTGATAATG GACCC; T7 promoter sequence underlined) and Rev-sens (TTAGGCCTGAGTTGAGTC) as

**Table 1. List of primers/probe for virus detection.**

| | Target | Name used in this study | Name | Sequence 5' to 3' |
|---|---|---|---|---|
| USA CDC, | N Gene | N1 | 2019-nCoV_N1-F | GACCCCAAAATCAGCGAAAT |
| | | | 2019-nCoV_N1-R | TCTGGTTACTGCCAGTTGAATCTG |
| | | | 2019-nCoV_N1-P | ACCCCGCATTACGTTTGGTGGACC |
| | N Gene | N2 | 2019-nCoV_N2-F | TTACAAACATTGGCCGCAAA |
| | | | 2019-nCoV_N2-R | GCGCGACATTCCGAAGAA |
| | | | 2019-nCoV_N2-P | ACAATTTGCCCCCAGCGCTTCAG |
| | N Gene | N3 | 2019-nCoV_N3-F | ATCACATTGGCACCCGCAATCCTG |
| | | | 2019-nCoV_N3-R | AGATTTGGACCTGCGAGCG |
| | | | 2019-nCoV_N3-P | TTCTGACCTGAAGGCTCTGCGCG |
| Charité, Germany | E gene | E | E_Sarbeco_F1 | ACAGGTACGTTAATAGTTAATAGCGT |
| | | | E_Sarbeco_R2 | ATATTGCAGCAGTACGCACACA |
| | | | E_Sarbeco_P1 | ACACTAGCCATCCTTACTGCGCTTCG |
| China, CDC | Orf1 / Rdrp gene | ORF1ab | ORF1ab-F | CCCTGTGGGTTTTACACTTAA |
| | | | ORF1ab-R | ACGATTGTGCATCAGCTGA |
| | | | ORF1ab-P | CCGTCTGCGGTATGTGGAAAGGTTATGG |
| Japan | N Gene | N | NIID_2019-nCOV_N_F2 | AAATTTTGGGGACCAGGAAC |
| | | | NIID_2019-nCOV_N_R2 | TGGCAGCTGTGTAGGTCAAC |
| | | | NIID_2019-nCOV_N_P2 | ATGTCGCGCATTGGCATGGA |
| Variants detection (this study) | Wild type S gene | Spike-F | Spike-F | TCAACTCAGGACTTGTTCTTAC |
| | | Spike-R | Spike-R | TGGTAGGACAGGGTTATCAAAC |
| | | WT-Spike | Wild type Spike-P | TGGTCCCAGAGACATGTATAGCAT |
| | Mutant S gene | Mutant spike 1 | Mutant S-Probe 1 | CCATTGGTCCCAGAGATAGCATGG |
| | | Mutant spike 2 | Mutant S-Probe 2 | CAGAGATAGCATGGAACCAAGTAA |
| | | Mutant spike 3 | Mutant S-probe 3 | CCCAGAGATAGCATGGAACCAAGT |
| | Orf1ab / Wild type NSP6 gene | NSP6-F | NSP6-F | CTAGTTGGGTGATGCGTATT |
| | | NSP6-R | NSP6-R | ACACAGTTCTTGCTGTCATAA |
| | | Wild type NSP6 | Wild type NSP6-P | AGTCTTTTAGCTTAAAACCAGACAAACTAGT |
| | Orf1ab / NSP6 mutant gene | Mutant NSP6 | Mutant NSP6 | AACACAGTCTTTTAGCTTCAAACTAGTATCAA |

TaqMan probes are labeled at the 5'-end with the reporter molecule 6-carboxyfluorescein (FAM) and with the quencher, Blackhole Quencher 1 (BHQ1) at the 3'-end.

**Table 2. List of primers/probe for cellular genes detection.**

| Target | Name used in this study | Name | Sequence 5' to 3' |
|---|---|---|---|
| RnaseP | RNP | CDC-RP-F | AGATTTGGACCTGCGAGCG |
| | | CDC-RP-R | GAGCGGCTGTCTCCACAAGT |
| | | CDC-RP-P | TTCTGACCTGAAGGCTCTGCGCG |
| TMPRSS2 | TMPRSS2 | TMPRSS2 Forward | TATAGCCTGCGGGGTCAAC |
| | | TMPRSS2 Reverse | CACTCGGGGGTGATGATGG |
| | | TMPRSS2 Probe | TCAAGCCGCCAGAGCAGGATCGT |
| ACE2 | ACE2 | ACE2 Forward | GGCTCCTTCTCAGCCTTGTT |
| | | ACE2 Reverse | GGTCTTCGGCTTCGTGGTTA |
| | | ACE2 Probe | TGCTGCTCAGTCCACCATTGAGG |
| IL1a | IL1a | IL1-a Forward | CATTGGCGTTTGAGTCAGCA |
| | | IL1-a Revrse | CATGGAGTGGGCCATAGCTT |
| | | IL1-a Probe | GTCAAGATGGCCAAAGTTCCAGACA |
| IL1b | IL1b | IL1b Forward | CAGAAGTACCTGAGCTCGCC |
| | | IL1b Reverse | AGATTCGTAGCTGGATGCCG |
| | | IL1b Probe | CCAGGACCTGGACCTCTGCCC |
| CXCL8 | CXCL8 | CXCL8 Forward | TGGACCCCAAGGAAAACTGG |
| | | CXCL8 Reverse | TTTGCTTGAAGTTTCACTGGCA |
| | | CXCL8 Probe | GTGCAGAGGGTTGTGGAGAAGTTT |
| IL6 | IL6 | IL6 Forward | TGCAATAACCACCCCTGACC |
| | | IL6 Reverse | GTGCCCATGCTACATTTGCC |
| | | IL6 Probe | TGCCAGCCTGCTGACGAAGC |
| IFNb1 | IFNb1 | IFNb1 Forward | AGTAGGCGACACTGTTCGTG |
| | | IFNb1 Reverse | GCCTCCCATTCAATTGCCAC |
| | | IFNb1 Probe | TGCTCTCCTGTTGTGCTTCTCCA |
| IFIT1 | IFIT1 | IFIT1 Forward | GATCTCAGAGGAGCCTGGCTAA |
| | | IFIT1 Reverse | TGATCATCACCATTTGTACTCATGG |
| | | IFIT1 Probe | CAAAACCCTGCAGAACGGCTGCC |

TaqMan probes are labeled at the 5'-end with the reporter molecule 6-carboxyfluorescein (FAM) and with the quencher, Blackhole Quencher 1 (BHQ1) at the 3'-end.

primers and the Q5-High Fidelity DNA polymerase (New England Biolabs, cat. no. M0491S). Thermal cycling conditions were: 98˚C for 30 s, 40 cycles of 98˚C for 10 s, 52˚C for 30 s, 72˚C for 1 min followed by 72˚C for 2 min (SimpliAmp Thermal Cycler, ThermoFisher Scientific). The size of the PCR product was verified on a 1.5% agarose gel in 0.5X TAE and further purified using the QIAquick PCR purification kit (Qiagen, cat. 28106). Template was used in the *in vitro* transcription reaction using the T7 RiboMAXExpress Large Scale Production System (Promega, cat. no. P1320) according to manufacturer's instructions. The *in vitro* transcript was extracted using phenol chloroform isoamyl followed by a chloroform wash and further precipitated using 300 mM sodium acetate in absolute ethanol at -20˚C, over-night. Sample was centrifugated at 4˚C for 30 min and washed twice with 70% cold ethanol. The RNA pellet was dried at RT for 5 min and resuspended in 100 µL water.

## Clinical samples

**Pulmonology department, Nice University Hospital.** 20 clinical samples from study participants were collected as part of the ELISpot study (ClinicalTrials.gov Identifier: NCT04418206). Participants were enrolled after signing written informed consent. Ethics committee approval was obtained from the "Comité de Protection des Personnes Sud

Méditerranée V" (registration # 2020-AO1050-39) on April 22, 2020. Nasopharyngeal swabs were collected in ESwab™ (COPAN) transport medium (2 mL), stored at 4˚C between sample collection and transport to the laboratory. One COVID-19 confirmed patient was collected consecutively using 2 different transport media: ESwab[TM] or a Tris-EDTA (TE: 10 mM Tris HCl pH 7.0, 2 mM EDTA, 20 μg/mL yeast tRNA) buffer.

**Garches Hospital (AP-HP).** Purified RNA samples (n = 92) from nasopharyngeal swabs with known COVID-19 status were obtained from the Garches Hospital following an ISO 15189 certified lab protocol previously described [19] and used to validate the Biomark[TM] HD protocol. Additional frozen VTMs from 55 patients with known COVID-19 status collected in 1–2 mL of saline buffer were obtained to set up the protocol. All samples were recorded for traceability on the basis of a unique barcode identifier.

**Bioesterel laboratories.** Samples (18 in virucide VTM, 2 in non-virucide VTM, Lingen) were stored overnight at 4˚C and processed the next day. Purified RNA samples (n = 74) from COVID-19 positive nasopharyngeal swabs with known status (collected week 3, 2021, in the region from Nice to Cannes, France) were used to screen for the presence of SARS-CoV-2 variants. A generic consent form was signed by the patients, allowing the utilization of the samples for scientific research. All samples were anonymized and their COVID-19 status was recorded.

## RNA extraction

Both miRNeasy Serum/Plasma Advanced Kit (Qiagen, cat. no. 217204) and QIAamp Viral RNA Mini Kit (Qiagen, cat. no. 52906) were used for Total RNA extraction from clinical samples according to manufacturer's instructions and using the Qiacube (Qiagen) apparatus. Final elution was performed in 20 μL water and 60 μL of AVE buffer for the miRNeasy Serum/Plasma Advanced Kit and QIAamp Viral RNA Mini Kit, respectively. Total RNA (containing miRNAs) were directly used for further analysis or stored at -20˚C.

## Detergent treatments of clinical samples

All clinical samples were handled in a biosafety level 2 laboratory and under biological safety cabinet using the adapted biosafety personal protective and respiratory equipment, according to the recommendations of the French society of microbiology (https://www.sfm-microbiologie.org/). The following detergents were used: Triton X-100 (Sigma, cat. no. T9284-100ML), Brij™35 (Millipore, cat. No. 1.01894.0100), Tween-20 (Sigma, cat. no. P7949-500ML), Brij™O10 (Sigma, cat. no. P6136-100G) and Poly Ethylene Glyco-600 (Aldrich, cat no. 202401-250G). They were included alone or in combination to prepare a 2X detergent master mix that was added to clinical transport media samples. A 10 mg/mL Proteinase K (Sigma P4850) solution was mixed extemporaneously with the 2X detergent master mix to reach a 2 mg/mL final concentration. Heating treatments were carried out at 95˚C for 5 min in a thermo-cycler apparatus (SimpliAmp Thermal Cycler, ThermoFisher Scientific). Treatment using the Quick Extract[TM] DNA Extraction Solution (Lucigen, QE09050) was performed using the same volume of reagent and sample, according to the manufacturer's instructions. Mixtures were then incubated at 95˚C for 5 min or at 60˚C for 10 min in a thermo-cycler. TE buffer was prepared using 10 mM Tris-HCl pH7.0, 2 mM EDTA, 20 μg/mL yeast tRNA with or without 0.5% Triton X-100.

## Detection of mRNA expression protocol using the Biomark[TM] HD system analysis

**Reverse transcription using the Fluidigm reverse transcriptase.** Extracted RNA (2 μL) or processed clinical samples were reverse transcribed using the Reverse Transcription Master

Mix kit according to the manufacturer's instructions (catalog number # PN 100–6297). Thermal cycling conditions were: 25˚C for 5 min, 42˚C for 30 min, 85˚C for 5 min.

**Reverse transcription using the Vilo SuperScript IV step.** 2.5 μL of processed clinical samples were used in a reverse transcription reaction using the SuperScript IV Vilo Master Mix (ThermoFisher Scientific, cat. no. 11756500) following the manufacturer's instructions. Reactions were incubated at 25˚C for 10 min, 55˚C for 10 min, 85˚C for 5 min.

**cDNA pre-amplification step.** 2.5 μL of cDNA were preamplified using the Preamp Master Mix kit (Fluidigm, cat. no. PN 100–5744) according to the manufacturer's instructions: 1 μL of Preamp Master Mix was combined with 1.25 μL of Pooled Taqman assay mix and 2.5 μL of cDNA in a 5 μL total volume reaction. The Pooled Taqman assay (180 nM) used in these reactions was prepared from an intermediate pooled Taqman assay solution (6.7 μM Forward and Reverse primers, 1.7 μM probes). Thermal cycling conditions were: 95˚C for 2 min followed by 20 cycles of 95˚C for 15 s, 60˚C for 2 min. After each preamplification reaction, samples were diluted 1:5 by adding nuclease-free water up to 25 μL.

**One step "RT-Preamplification".** One step "RT-Preamplification reactions" were performed using the Cells Direct One-Step qRT-PCR kit (Invitrogen, cat. no. 46–7200) according to a protocol adapted from the Reverse Transcription-Specific Target Amplification (Fluidigm). Briefly 5.5 μL of total RNA were mixed with 6.25 μL the Pooled Taqman assay, 0.5 μL Superscript III RT/Platinum Taq, 12.5 μL 2x Reaction mix and nuclease-free water to 25 μL total volume. Thermal cycling conditions were: 42˚C for 15 min, 95˚C for 2 min followed by 15 to 20 cycles of 95˚C for 15 s, 60˚C for 2 min. Then, samples were diluted 1:5 using nuclease-free water up to 125 μL.

**Real time qPCR using Biomark^TM HD system.** PCR was performed following Gene Expression Standard TaqMan Assays protocol (Fluidigm cat n˚ 100–6170 C1), using a 10X assays mix and a pre-sample mix prepared separately. The 10x assays mix was prepared by mixing 2 μL of combined Primer (Forward/Reverse 6.7uM, Probe 1.7 μM) and 2ul 2X Assay Loading Reagent (Fluidigm PN 100–7611) to a final volume of 4 μL (per reaction). The pre-sample mix was prepared by mixing 2 μL TaqMan Universal PCR Master Mix (2X) (Life Technologies PN 4304437) and 0.2 μL 20X GE Sample Loading Reagent (Fluidigm PN 100–7610) and 1.8 μL preamplified cDNA to a final volume of 4 μL.

Then, 3 μL of 10x assays mix and of pre-sample mix are transferred into the 192.24 IFC, loaded into the Biomark^TM IFC controller RX and transferred to the Biomark^TM HD apparatus. Thermal cycling conditions were as follows: 50˚C for 120 s, 95˚C for 600 s followed by 20 cycles of 95˚C for 15 s, 60˚C for 1min.

## Detection of microRNA-expression protocol using the Biomark HD system analysis

RNAs (1 μL) were processed using the miScript 2 RT kit (QIAGEN cat n˚ 218161). Preamplification was performed using 5 μL of diluted cDNA using the mi Script microfluidics PreAMP kit (QIAGEN, cat n˚ 331455) and following the 384-plex preamplification cycling conditions (see manufacturer's protocol). QPCR was performed following miScript microfluidics PCR kit (QIAGEN cat n˚ 331431) with commercially-available primers (miScript Primer assays). Six μL of assays mix and sample mix were prepared as described in the manufacturer's protocol and 5 μL of each are loaded onto the 96.96 IFC plates (Fluidigm cat n˚ BMK-M-96.96). The list of microRNAs (miRNAs) tested in this study is presented in S1 Table.

## Biomark HD system analysis

Exhaustive protocols are available at:

https://dx.doi.org/10.17504/protocols.io.bd3ii8ke (Two-Step qPCR Protocol)
https://dx.doi.org/10.17504/protocols.io.bnx4mfqw (One-Step qPCR Protocol)

### Fluidigm HD system qPCR analysis

qPCR results were analyzed using the Real Time PCR analysis software provided by Fluidigm.

### Statistical analysis

Statistical analyses were performed using GraphPad Prism software. Results are given as mean ± SEM. One-way ANOVA followed by Kruskal-Wallis test was used for multiple comparison. P-value $< 0.05$ was considered significant.

## Results

### Performance and validation of a Biomark-HD SARS-CoV-2 qPCR assay

We first set up a classical Fluidigm qPCR assay protocol that combines all primers/probe sets in a single reaction. This system requires a pre-amplification step after cDNA synthesis. This is due to the low volume of the IFC reaction chamber (9nl). As a result, and compared to the traditional RT-qPCR protocol (cDNA synthesis and qPCR), three consecutive reaction steps (RT, pre-amplification and qPCR) are required in the classical Biomark[TM] -HD assay.

To evaluate the efficiency of the qPCR, we first assessed the test sensitivity using a range of synthetic viral N transcript dilutions with the US CDC primers/probe set N1. We found that the N transcript detection limit was seven transcript copies per reaction (Fig 1A). Linear regression showed a good correlation across Cq value and dilution series of N transcript ($R^2 = 0.9846$) (Fig 1A). Similar sensitivity experiments were performed with additional primers/probes from the US CDC and by diluting the transcript in TE buffer without (S1A Fig) or with a RNA purification step (S1B Fig). Dilution of a SARS-CoV-2 positive clinical sample also indicated a large dynamic range of detection of 6 orders of magnitude for N1 and the E [20] primers/probe (Fig 1B). We then tested the combination of various primers/probe sets during the pre-amplification reaction, showing that 2 distinct mixes (mix #1: N1, E, human RNP; mix #2: ORF1ab, E, N and human RNP) gave similar data on clinical samples (Fig 1C). We finally evaluated the performance of the test on a set of 18 samples from positive patients showing a large range of Cq values. The initial reference analysis was performed using the Elitech Genefinder[TM] COVID-19 detection kit, comprising 3 sets of viral primers/probe targeting the same viral genes as the mix #2. We noticed excellent correlations between both methods for the detection of the 3 transcripts with $R^2$ above 0.95 (Fig 1D), only affected by some high-range samples (i.e. Cq below 5), which displayed a saturation on Biomark[TM]. Similarly, we also evaluated the performance of the test using the probe of the viral N gene on a set of 92 purified RNA samples of known status, comprising 15 positive samples, previously analyzed using the GeneFirst COVID-19 detection kit (comprising two sets of primers/probe including the N target). We plotted the correlation for the Cq obtained for the detection of the N transcript using both methods, showing a $R^2$ of 0.97, with no false positive detected on the remaining 77 negative samples (S2 Fig).

### Optimization of the test

Additional tests were performed to reduce handling and reaction times by optimizing cycle numbers, elongation times and primer concentrations in the preamplification reaction (S3 Fig). In another optimization, we combined RT and pre-amplification into a one-step reaction, using the Cells Direct One-Step qRT-PCR kit (ThermoFisher Scientific). We compared this method to the classical Fluidigm RT-qPCR on a set of 18 clinical samples with a wide range of

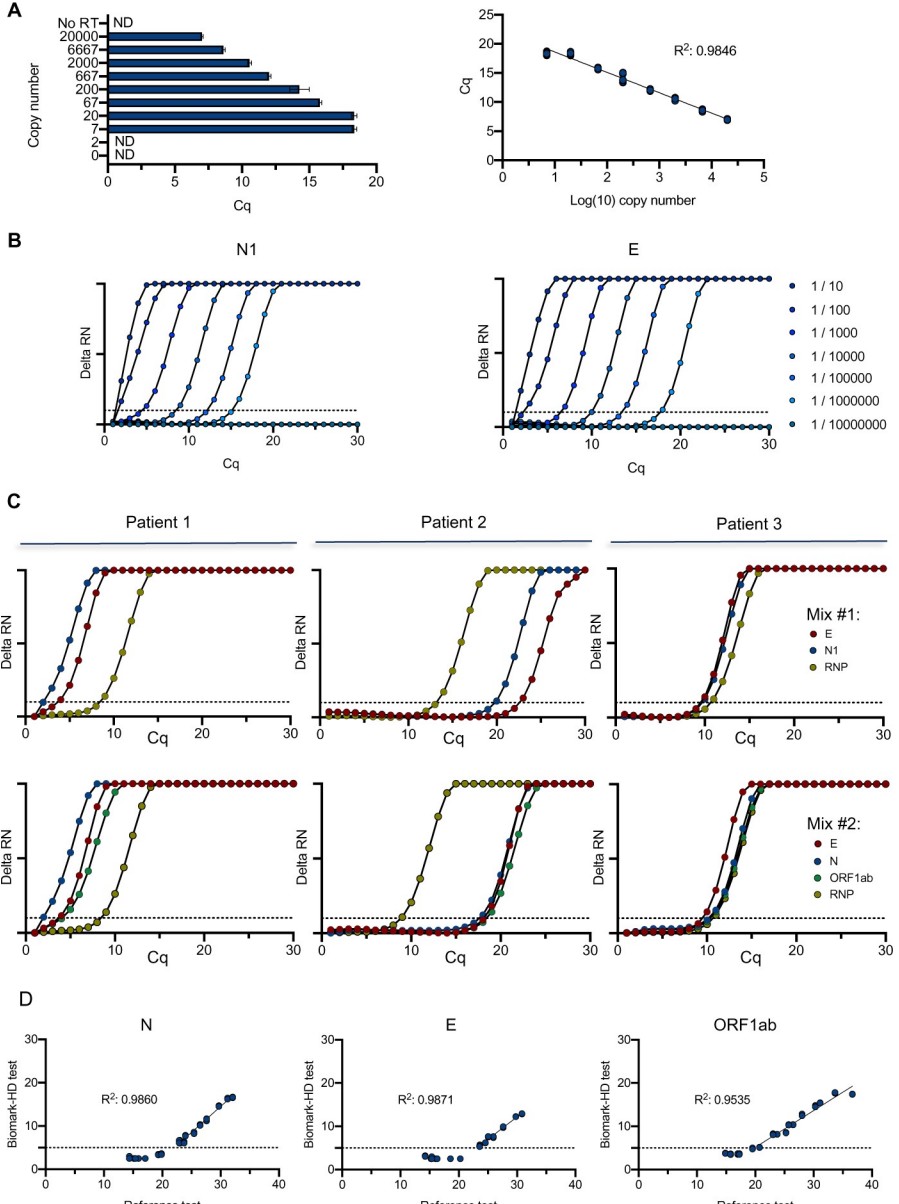

**Fig 1. Performance and validation of the Biomark-HD SARS-CoV-2 qPCR assay.** (A) Titration of the diluted (N) nucleocapside spike-in transcript. A serial dilution of the synthetic N transcript from $2.10^4$ to 2 copies was performed and processed through the Biomark-HD protocol. Correlation and amplification curves of detected Cq values according to synthetic N transcript copy number obtained with N1 primers/probe are shown. (B) Amplification curves showing the range of detection of a SARS-CoV-2 positive clinical sample serial dilution with N1 and E primers/probe. (C) Typical amplification curves showing the combination of two primers/probe sets on 3 SARS-CoV-2 positive clinical samples. Mix #1: E, N1, human RNP. Mix #2: E, N, ORF1ab and human RNP. (D) Validation of the Biomark-HD protocol on a cohort of 18 biopsies from positive patients. The correlation of the Cq values obtained for the N, E and the ORF1ab genes are presented. The data presented are representative of at least two independent experiments performed in quadruplicate.

Cq values. Very similar Cq values were obtained with both methods for the cellular (RNP) and for 2 viral (E and N) primers/probe sets. The detection limit for ORF1ab gene was even improved using the Cells Direct One-Step qRT-PCR kit-based protocol (Fig 2A). Overall, this protocol offers the possibility to run 192 samples with 24 couples of primers/probe sets (using the Fluidigm 192.24 IFC) corresponding to 4608 end-points in less than 3 hours.

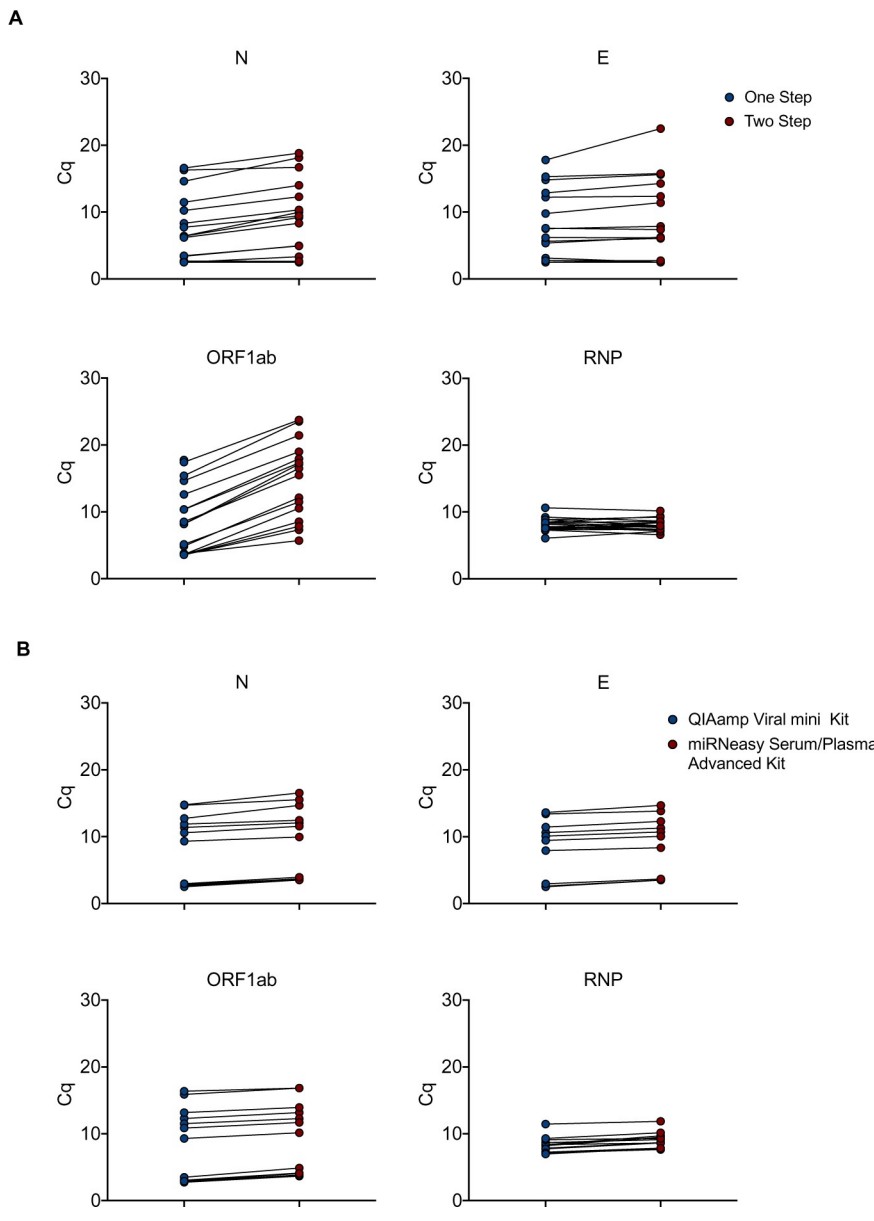

**Fig 2. Optimization of the assay.** (A) Total RNA from 18 clinical samples with a wide range of SARS-CoV-2 infection were subjected to either a Two-step reaction (red circles; consecutive Reverse Transcription and Pre-amplification) or One-step reaction (blue circles; combined Reverse Transcription and Pre-amplification). Quantitative PCR reactions were performed on the Biomark-HD using cellular (RNP) and viral (N, E, ORF1ab) primers/probe sets. (B) Total RNA from 18 clinical samples with a wide range of SARS-CoV-2 infection were extracted with either the miRNeasy Advanced Serum Plasma Kit (red circles) or the Virus QIAamp Viral RNA kit (blue circles). RNAs were processed using the One step reaction. Quantitative PCR reactions were performed on the Biomark-HD using cellular (RNP) and viral (N, E, ORF1ab) primers/probe sets. The Cq presented are representative of two independent experiments performed in quadruplicate.

## Alternative RNA extraction method using the miRNeasy Serum/Plasma Advanced Kit (Qiagen)

To limit the possible impact of a shortage of viral RNA extraction kits, we compared the performance of the QIAamp Viral mini kit to that of the miRNeasy Serum/Plasma Advanced Kit.

A set of 18 SARS-CoV-2 positive samples with a wide range of Cq values was extracted using both protocols. The 36 resulting RNA samples were run in parallel on the BIOMARK-HD using 3 human and 3 viral primers/probe sets. Identical Cq values were obtained for the human probe (RNP) with both extraction protocols. A slight gain of signal (around 1–2 Cq) was measured after QIAamp Viral purification for the 3 viral primer/probes (E, N and ORF1ab) (Fig 2B). These data suggest that the QIAamp Viral RNA mini Kit slightly increases the efficiency of viral transcript extraction, but the miRNeasy Serum/Plasma Advanced Kit represents an acceptable alternative solution in case of shortage.

## Use of the Biomark™ HD-based protocol to analyze the host response to SARS-CoV-2 infection at the mRNA and miRNA levels

Exploiting the potential for multiplexing of the Biomark™ HD-based technology, we included sets of human primers/probe to evaluate the expression of genes required for viral entry (*ACE2*, *TMPRSS2*), as well as inflammatory/antiviral response genes (*CXCL8*, *IL1A*, *IL1B*, *IL6*, *IFNB1*, *IFIT1*). A set of miRNAs (see S1 Table for detail) was also tested (Fig 3A). We divided the patient samples (n = 72) into groups, according to their viral load (strong, medium, weak or negative). *ACE2* and *TMPRSS2* were detected in all samples, regardless of the viral status, with Cq values close to those for *RNP*, suggesting they could represent valid human control genes to assess the quality and the presence of epithelial cells in the samples (Fig 3B and 3C). The signal obtained for the inflammation and interferon responses was more heterogeneous (Fig 3B). We particularly noticed a strong signal (in the range of 5 Cq) for *IFIT1* in the 3 samples with the highest viral load (Cq values < 10 for the viral probes) (Fig 3C). Additionally, we evaluated the possibility to quantify miRNA levels in the same samples. A large number of miRNAs were detected in most of the samples (examples in Fig 3B and 3C). Overall, our results demonstrate that cellular markers can be easily quantified in nasopharyngeal swabs and may provide useful information to refine the diagnosis or prognosis of COVID-19 patients.

## Detection of mutations associated with SARS-CoV-2 variants by RT-qPCR

We next assessed whether the potential of the Biomark™ HD-based technology for multiplexing was adapted to the detection of SARS-CoV-2 variant in patient samples. For this purpose, a set of probes was designed to detect selectively SARS-CoV-2 genomes bearing the H69-V70 deletion (21765–21770) which emerged in several lineages of the virus, including the B.1.1.7/501Y.V1 variant and the 3675-GSF deletion (11288–11296) detected in all 501Y variants (B.1.1.7/501Y.V1, B.1.351/501Y.V2, P.1/501Y.V3) [21] (Fig 4). In a set of 74 COVID-19 patient samples tested in mid-January in the south of France, 11 samples showed loss of signal for wild-type spike and NSP6 probes while all the corresponding variant probes were detected, strongly suggesting the presence of the B.1.1.7 variant in these 11 samples. This experiment demonstrated that this methodology can be rapidly adapted according to SARS-CoV-2 evolution, allowing in a single assay the COVID-19 diagnosis as well as information regarding the presence of various variants of interest.

## Direct qPCR detection of SARS-CoV-2 using inactivating lysis buffers-based protocols

Direct SARS-CoV-2 diagnostic methods have been described, mainly performing RT-qPCR directly on crude or heat-inactivated subject samples [8–10]. An additional refinement would be to improve the sensitivity and safety of these protocols by the use of RT-qPCR-compatible inactivating lysis buffers containing detergents such as Triton X-100, a standard non-ionic

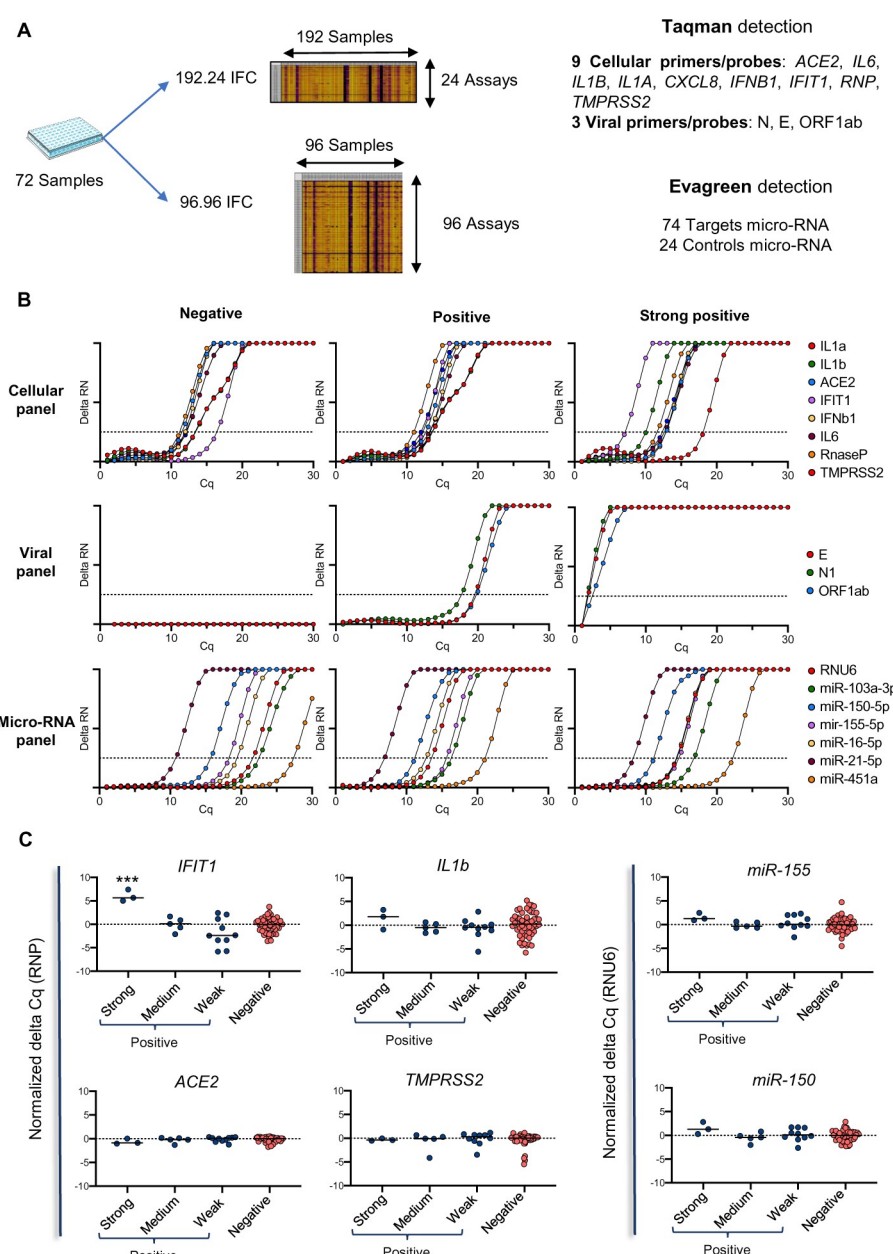

**Fig 3. Use of the Biomark-based protocol to analyze the host response to SARS-CoV-2 infection at the mRNA and microRNA levels.** (A) Overview of the analysis strategy on 72 patient samples. They were divided into 4 groups, according to their viral load, from negative, weak (Cq for viral probes >20), medium (20 > Cq for viral probes >10) and strong (Cq for viral probes < 10) SARS-CoV-2 positive. (B) Typical amplification curves of the different genes (cellular, viral and micro-RNA) on three different SARS-CoV-2 patient status. (C) Modulation of cellular markers in the different groups of patients according to their SARS-CoV-2 viral load. IFIT1 expression was statistically elevated (p < 0.001) in the strong COVID-19-positive samples compared to the three other groups considered separately. The Cq presented are representative of two independent experiments performed in quadruplicate.

detergent widely used for inactivation of enveloped viruses such as SARS-CoV [22, 23]. We first compared a regular extraction-based method with a direct RT-qPCR protocol on a synthetic N transcript spike in TE buffer in the presence or the absence of 0.5% Triton X-100 and confirmed that Triton X-100 did not interfere with the efficiency of the RT-qPCR steps (S4A

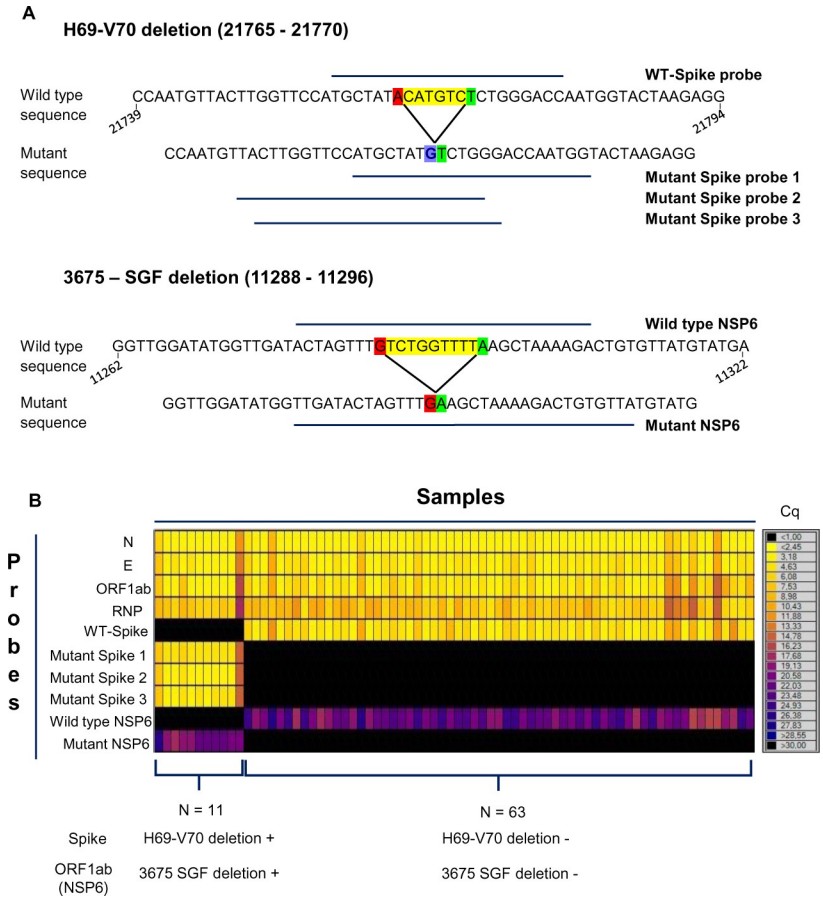

**Fig 4. Use of the Biomark-based protocol to detect mutations associated with SARS-CoV-2 variants.** (A) Overview of the probes design used for the detection of two deletion mutations found in the Spike gene (H69-V70, top panel) and in the NSP6 gene (3675-SGF, bottom panel). Deletions are highlighted in yellow with both nucleotide at the 5' and 3' side shown in red and green respectively. Of note: for the H69-V70 mutation an A-to-G punctual deletion is shown in blue. Three mutant probes and one wild type probe were generated for the H69—V70 deletion (21765–21770) and one mutant and one wild type probes were generated for the 3675 –GSF deletion (11288–11296). The nucleotide numbering is based on the reference SARS-CoV-2 complete sequence (NCBI Reference Sequence: NC_045512.2). (B) Heatmap of Cq value from RT-qPCR using various combinations of primer/probes performed on 74 SARS-CoV-2 positive clinical samples. Cq scale range is shown on the right. Using the Biomark-based protocol, 11 positive samples for the mutation associated with the SARS-CoV-2 variants have been detected (on 74 positive clinical samples).

Fig). We then performed two consecutive nasopharyngeal swab samplings on a COVID-19 diagnosed patient, using either a regular VTM or a TE buffer containing 0.5% Triton X-100 followed or not by a heating step at 65˚C for 10 min. As expected, the presence of Triton X-100 and the heating process did not affect the detection of the human RNP transcript for both RT-qPCR methods. Conversely, while the Triton X-100-lysis buffer/heating process slightly improved the signal for the virus N1 primers/probe using the regular extraction-based protocol, this treatment strongly inhibited the direct RT-qPCR method for the same cellular transcript (S4B Fig), indicating that 0.5% Triton X-100-lysis buffers are not compatible for a sensitive direct SARS-CoV-2 RT-qPCR assay.

We then tested the use of additional detergents and emulsifiers (Tween-20, Brij™-35, Brij™ O10), alone or in combination with polyethylene glycol (PEG600), in the presence or the absence of a treatment with proteinase K (PK), followed by heat inactivation. The experiment was performed on 2 clinical samples of known COVID-19 status, sampled in a commercial

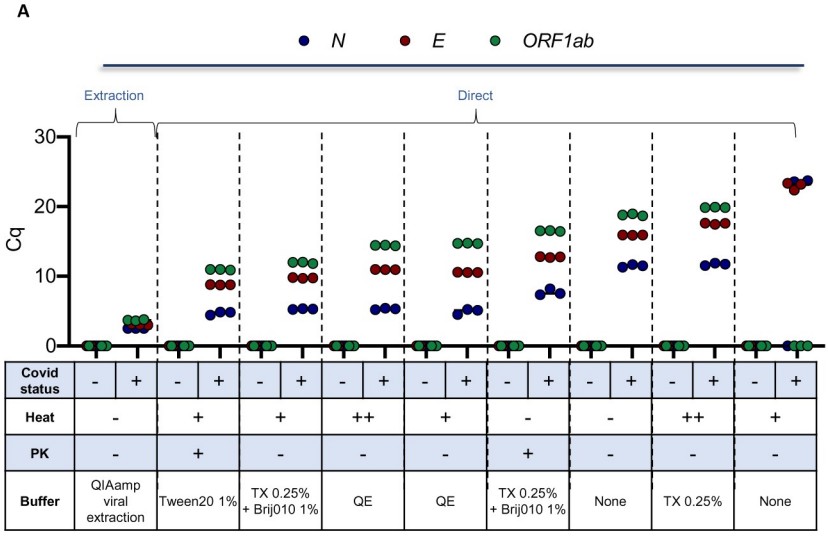

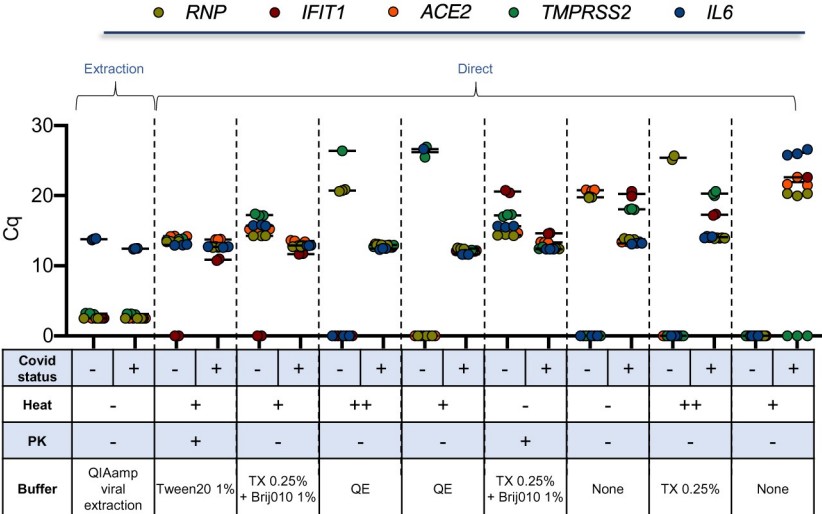

**Fig 5. Detection of viral and cellular genes using inactivating lysis buffers-based protocols.** Two clinical samples of known COVID-19 status (one positive and one negative) sampled in a commercial VTM medium, were aliquoted and then treated in parallel using the indicated detergent conditions (selected from S5 Fig) in the presence or absence of PK. RNA was extracted as a control. Biomark-HD RT-qPCR was performed using (A) viral (N, E, ORF1ab) and (B) cellular (ACE2, IFIT1, IL6, TMPRSS and RNP) primers/probe sets. Legend: +: 10 min at 65°C; ++: 10 min at 65°and 5 min at 95°C; TX: Triton X-100; QE: Quick Extract™ DNA Extraction Solution. The Cq presented are representative of two independent experiments performed in quadruplicate.

VTM medium, aliquoted and then treated in parallel using 16 distinct detergent conditions in the presence or absence of PK, for a total of 64 experimental conditions (Fig 5 and S5 Fig). Additional controls included the crude and heat-inactivated samples, the use of Quick Extract™ DNA Extraction Solution (QE), which has been recently proposed as an alternative method to extraction [24, 25] as well as a regular extraction-based protocol (positive control). Three and five sets of SARS-CoV-2 (Fig 5A) and human (Fig 5B) primers/probe, respectively, were used in the same assay on a 192.24 dynamic array. In these conditions, a direct assay on a

crude sample resulted in a 9 to 15-Cq increase, depending on the viral primers/probe set, when compared to the extraction method, with an even more pronounced effect when samples were heated 5 min at 95˚C (Fig 5A). While the addition of 0.5% Triton X-100 alone was similar to the crude direct assay condition, the combination of Triton X-100 with the emulsifier Brij™ O10 (Oleth-10) improved the detection for all viral primers/probes, with a drop of around 5 Cq compared to Triton X-100 alone. All other detergents used alone (Tween-20, Brij™ O10, Brij™-35) gave a similar signal as 0.5% Triton X-100 alone. Of note, the addition of PK resulted in an improvement of sensitivity in the presence of various detergents including Tween-20, Brij™ O10 and Brij™-35 while addition of PEG had no beneficial effect (Fig 5 and S5 Fig).

Based on these results, we selected two of the best direct detergents based assays (Triton X-100/Brij™ O10 and Tween-20/PK) and compared the sensitivity of these methods with a crude direct assay and the extraction method on additional clinical samples. For all direct assays, all samples were heat-inactivated at 95˚C (5 min). Moreover, we used a set of 17 clinical samples from SARS-CoV-2-diagnosed patients collected in saline solution with a wide range of Cq values. To control pH conditions and limit RNA degradation, all samples were diluted in TE buffer. We plotted the relationships between Cq obtained for the detection of the N and ORF1ab primers/probe sets for the Biomark assay and the GeneFirst COVID-19 detection kit (Fig 6). As expected, an excellent correlation was obtained when comparing the RNA extraction-based Fluidigm protocol with the reference assay ($R^2$ = 0.8258 and $R^2$ = 0.8208, all samples detected). Conversely to the data obtained with the commercial VTM, the best direct protocol corresponded to the crude assay (TE), approaching a very similar performance to that obtained with the extraction-based assay ($R^2$ = 0.6447 and $R^2$ = 0.6782, 16/17 samples detected). The $R^2$ values dropped in a dramatic way for the two direct detergents based assays, with 11 to 14 positive samples detected out of 17. Overall, these data suggest that a direct qPCR method using saline as a VTM and a basic TE buffer followed by a 5 min inactivation step at 95˚C can efficiently support detection of SARS-CoV-2.

## Discussion

The present paper describes a reliable and flexible multiplex nanofluidic qPCR system-based protocol to detect SARS-CoV-2. Its versatile format makes it easily adaptable to detect mutations associated with SARS-CoV-2 variants, multiple other pathogens and/or host cellular markers. The same run allows to test in parallel viral and host RNAs, including miRNAs. We show a high concordance between this method and clinically approved traditional qPCR tests. This assay addressed some of the challenges of RT-qPCR assays, including analyzing a larger number of reactions per run, making the assay more cost-effective and less time-consuming. Further, IFCs dynamic arrays not only reduces the reaction volume from about 10 µL down to the 10 nL scale, but allows large multiplexing as well as increased parallelization throughput of qPCR reactions.

The flexibility of this platform may be used in biomarker studies aiming at predicting at diagnosis the severity (requirement or not for hospitalization/intensive care) or the length (development of so-called "long-COVID") of the disease. Several studies have depicted an elevated innate and adaptive immune activation in severe COVID-19 patients [26–28], and a differential immune phenotype in moderate versus severe disease after the second week of infection [28, 29]. The Biomark HD offers the possibility to extend the number of genes tested to biomarkers including pro-inflammatory cytokines, chemokines, interferons, tissue repair genes and miRNAs by nasopharyngeal swabs or in saliva.

Monitoring miRNAs expression levels in swabs or saliva during the course of infection might be of great interest, because of their stability in biological fluids and frequent alterations

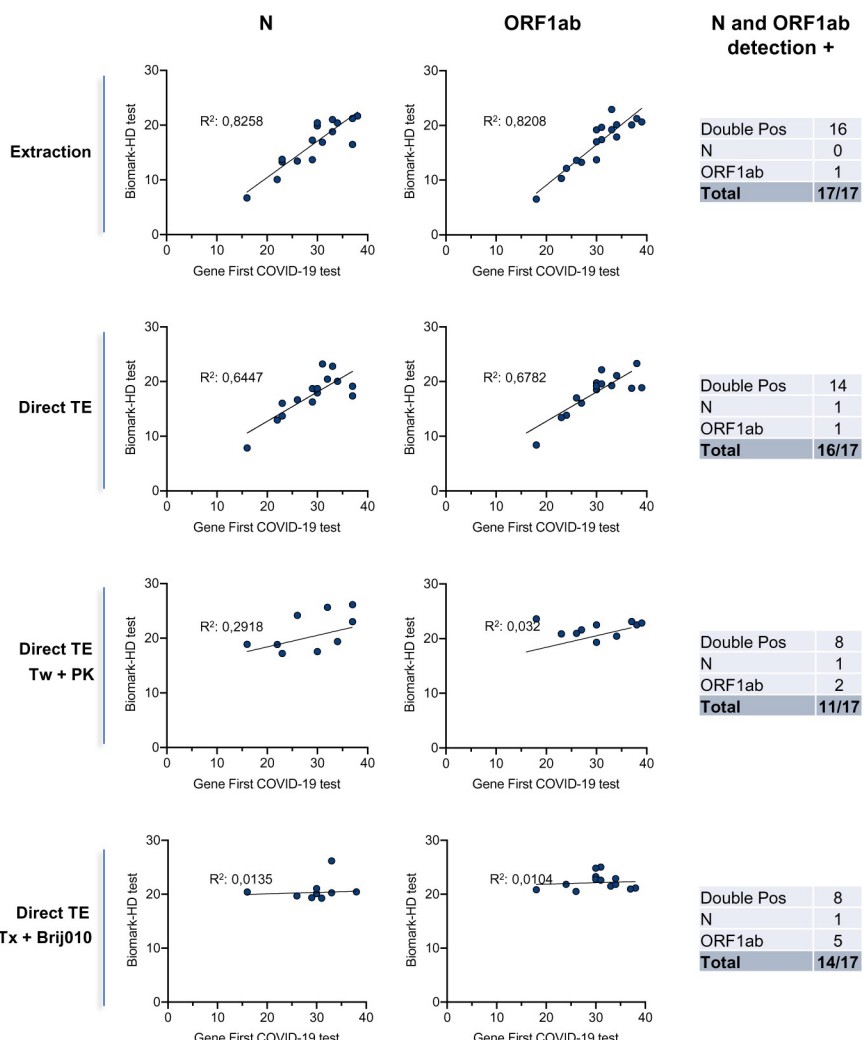

**Fig 6. Correlation of the detection of two viral genes (N, ORF1ab) obtained using a commercial COVID-19 detection kit and extraction-based or direct Biomark-HD assays.** We used a set of 17 clinical samples from SARS-CoV-2-diagnosed patients collected in saline solution with a wide range of Cq values. To control pH conditions and limit RNA degradation, all samples were diluted 2X in TE buffer. RNA was extracted (as a control) or samples were treated using Tween 20 and PK (Tw + PK) or Triton X-100 and Brij010 detergent solutions (Tx + Brij010). The different RT-qPCR Biomark methods were compared with the GeneFirst[TM] COVID-19 method as a reference. The correlation between the Cq obtained with the two methods is presented. One negative sample was processed in the three direct conditions (TE, TE plus Tw + PK and TE plus Tx + Brij010) with as results an absence of signal with the two probes (Cq >40). Double Pos: double positive.

during both chronic and acute airway infections [30, 31]. For instance, small RNA profiling of *in vitro* SARS-CoV-2 infected cell lines resulted in an increased expression of miR-155, one of the miRNAs detected in our clinical samples (Fig 3C) [32]. This miRNA, frequently associated with virus infection, was recently shown to contribute to the development of lethal acute respiratory distress syndrome (ARDS) in H1N1 influenza A virus-infected mice, suggesting it could represent a valuable biomarker to follow the individual immune response of patients [33]. Additionally, several other miRNAs targeting *ACE2* and *TMPRSS2* might influence SARS-CoV-2 entry and the course of the disease, as recently suggested [34].

With the emergence of new SARS-CoV-2 variants that are potentially associated with increased transmissibility, immune escape and/or vaccine efficacy reduction [35–37], it is

crucial to make available a test that determines whether a patient is SARS-CoV-2-positive but also identified the variant type. We demonstrate here that the Biomark[TM] HD-platform is particularly adapted to detect the presence of known variants in patient samples. Such a methodology, which is more cost-effective and flexible than sequencing, is likely to have implication in the epidemiological monitoring of the pandemic.

An important limitation of current tests is the requirement for RNA extraction that constitutes an obstacle to scale-up the capacity of testing both in term of time and cost. Several groups have explored methods to circumvent RNA extraction by performing RT-qPCR directly on crude or heat-inactivated clinical samples [8–10]. Overall these methods show that testing for SARS-CoV-2 infection can be performed without RNA extraction, with a limited loss in accuracy for determining negative and positive cases. While this procedure is simple and attractive, it might be improved by the addition of detergents to facilitate viral capsid lysis to release genomic RNA and also directly inactivate the virus to facilitate sample handling and safety. Several methods and commercial kits have developed approaches to lyse efficiently mammalian cells and directly perform RT-qPCR or RNA-seq libraries. However, these methods are not fully optimized for virus lysis, which requires increased concentrations of detergents. Some studies indicate that Triton X-100, widely used in virus inactivation procedures, or Tween-20 may slightly improve or at least not interfere with the RT-qPCR SARS-CoV-2 direct testing of nasopharyngeal swabs or saliva [9, 16]. However, these initial reports clearly mentioned that additional efforts were needed to optimize direct RT-qPCR assays on detergent-inactivated samples. We evaluated here the use of several detergents and emulsifiers (Triton X-100, Tween-20, Brij™-35, Brij™ O10), alone or in combination, in the presence or the absence of a treatment with PK or polyethylene glycol (PEG) followed by heat inactivation to assess their compatibility with a direct reverse transcription enzymatic step. This initial screening indicated that the addition of each of these detergents, alone, did not improve the assay compared to a direct assay performed on a crude commercial VTM sample. By contrast, the addition of PK improved the detection with all the detergents tested, in agreement with a recent study [38]. Of note, a combination of Triton X-100 with Brij™ O10, in the absence of PK, had a similar effect, by decreasing the Cqs for the 3 SARS-CoV-2 primers/probes. (Fig 4A) Brij™ O10 contains Oleth-10, a polyoxyethylene oleyl ether used in aqueous emulsions that may increase solubilization of the viral capsid or contribute to stabilization of the emulsion. When we tested the three best direct protocols on a cohort of samples collected in saline solution, we could not confirm the detection improvement observed in the screening assay using both lysis buffers (Tween-20/PK and Triton X-100/Brij™ O10, Fig 5). This discrepancy may arise from the fact that the screened samples were collected in a commercial VTM while the cohort experiment was performed on biopsies collected in saline buffer. This difference suggests a complex interaction between the various constituents of this VTM (pH, nature of the medium, presence of albumin, gelatin, anti-bacterial agents, for examples), and the different detergents and emulsifiers tested. Further developments are definitely required to fine tune specific combinations of detergents and emulsifiers with specific VTMs. In any case, our data clearly indicate that a direct qPCR method using saline as a VTM and a basic TE buffer followed by a 5 min inactivation step at 95°C shows almost identical performance to that of a classical extraction-based assay, allowing notably the identification of samples with high Cq values (Fig 5). Considering that SARS-CoV-2 remains detectable in phosphate buffer for up to a month when stored at various temperatures [39], our data supports the use of PBS, as an alternative to VTM for direct SARS-CoV-2 testing. We further confirm and refine previous studies showing that testing for SARS-CoV-2 infection can be performed with simplified protocols omitting RNA extraction steps without major loss in accuracy [8, 9]. Based on the data presented here, the simplest sample collection and preparation for direct RT-qPCR COVID-

19 test would be to sample the swab into a small volume (around 0.5 mL) of TE buffer at a pH of 7.0 to limit dilution of the virus, supplemented with yeast tRNA or RNAse inhibitors to increase RNA stability. This type of buffer was recently shown to have an excellent capacity to preserve the SARS-CoV-2 signal [9, 39]. Lysis buffers containing detergents would provide the possibility to fully inactivate the virus, allowing a rapid and safe handling of the clinical samples, but our data indicate that their efficiency may vary depending on clinical sample processing and should be further optimized.

In conclusion, we propose that such a direct qPCR procedure offers an interesting option for massively scaling up SARS-CoV-2 testing. We believe that this protocol can be particularly well-adapted for pooling approaches [40], to screen asymptomatic individuals in communities at risk. It would provide an additional tool to enhance testing capacity and affordability across the world, as widely recommended by the Health Community worldwide, such as the "all-in" approach to testing recently proposed [4].

## Supporting information

**S1 Fig. Titration of the diluted nucleocapsid spike-in transcript.** A ten-fold serial dilution ranging from 1 to 10–6 was prepared from the stock solution of the in vitro-transcribed N gene and supplemented with 2 ng/μL of total RNA from HEK 293 Cells. Reverse Transcription was performed followed by 15 cycles of pre-Amplification and 30 cycles of qPCR. The RT-qPCR reaction was performed without (A) or with (B) a RNA purification step. Linear regression was performed by logarithmic plots of transcript copy number against Cq value. We observed a good correlation according Cq linear regression curves according to dilution for the three viral CDC primers/probe sets (N1, N2, N3). No Cq value has been detected for the E primers/probe. RNP, used as internal control, shows constant detection of Cq value, suggesting a good performance of the qPCR.
(TIF)

**S2 Fig. Validation of the Biomark[TM] -HD protocol on a cohort of 92 biopsies including 15 positive patients.** The correlation of the Cq values obtained for the N primers/probe (Biomark[TM] -HD) and the GeneFirst COVID-19 detection kit is presented.
(TIF)

**S3 Fig. Preamplification step optimization.** The elongation time used in the preamplification reaction was reduced to 1 min at 60˚C (A) from 2 min at 60˚C (B) using diluted total RNA from a SARS-nCov2 positive patient sample.
(TIF)

**S4 Fig. Effect of Triton X-100 on extraction-based and direct RT-qPCR protocol performed on a synthetic N transcript or a clinical sample.** A. In vitro-transcribed viral N gene was added to either a Triton X-100 containing lysis buffer (Tx) or to TE buffer (TE). The samples were heated or not at 65˚C for 10 min. RNA extraction was performed or not (direct) and N1 or RNP levels were determined by RT-qPCR using the Biomark[TM] -HD system. B. A similar protocol as in A was used but the starting material were two consecutive sample collection from the same patient processed either in a Triton X-100 containing lysis buffer or to TE buffer.
(TIF)

**S5 Fig. Heatmap of Cq values from direct RT-qPCR experiments using various combinations of detergent on clinical samples.** Samples from a positive- or negative-COVID-19 patient collected in a commercial VTM was mixed with different combinations of detergents/

emulsifiers, in presence or absence of PK (2 mg/mL) and further heat at 95˚C for 5 min or not. Cq values obtained in quadruplicate are presented. Tx: Triton X100; PEG: poly ethylene glycol 600; QE: Quick Extract$^{TM}$ DNA Extraction Solution.
(TIF)

**S1 Table. List of the micro-RNA tested using the Biomark HD system.**
(DOCX)

## Acknowledgments

We thank our colleagues from the IPMC and the Délégation CNRS Côte d'Azur for their help and support, especially Catherine Lecalvez, Michel Bordes, Véronique Campbell and Simon Szmidt for administrative support and gratefully acknowledge the staff from the Département de Pneumologie of the Nice Hospital, especially Jennifer Griffonet and Charlotte Maniel. We also thank the technical support of the UCA GenomiX platform and MICA imaging facility of the University Côte d'Azur. We are grateful to Bayer SAS for loan of equipment and reagents, to Hervé Groux (Immunosearch) as well as to Dr Pol-Henri Guivarch and Alexandre Romain (Agence Régionale De Santé PACA) for their support and interesting discussion. We are also grateful to the "Propagate" consortium, notably Syril D. Pettit, Emily A. Bruce, Jason W Botten and Keith R. Jerome for their support and fruitful discussion.

## Author Contributions

**Conceptualization:** Julien Fassy, Caroline Lacoux, David Rouquié, Pascal Barbry, Laure-Emmanuelle Zaragosi, Bernard Mari.

**Formal analysis:** Bernard Mari.

**Funding acquisition:** Jean-Louis Nahon, Bernard Mari.

**Investigation:** Julien Fassy, Caroline Lacoux, Sylvie Leroy, Latifa Noussair, Aurélien Degoutte, Georges Vassaux, Charles-Hugo Marquette, Pascal Barbry, Laure-Emmanuelle Zaragosi, Bernard Mari.

**Methodology:** Julien Fassy, Caroline Lacoux, Sylvain Hubac, Patrick Touron, Antoinette Lemoine, Jean-Louis Herrmann, Laure-Emmanuelle Zaragosi, Bernard Mari.

**Project administration:** Jean-Louis Nahon.

**Resources:** Sylvie Leroy, Latifa Noussair, Sylvain Hubac, Vianney Leclercq, David Rouquié, Charles-Hugo Marquette, Martin Rottman, Patrick Touron, Antoinette Lemoine, Jean-Louis Herrmann.

**Supervision:** Bernard Mari.

**Writing – original draft:** Julien Fassy, Caroline Lacoux, Laure-Emmanuelle Zaragosi.

**Writing – review & editing:** Georges Vassaux, Pascal Barbry, Bernard Mari.

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
