## [Decision Letter · Decision Letter 0]

22 Jan 2021

PONE-D-20-36827

Versatile and flexible microfluidic qPCR test for high-throughput SARS-CoV-2 and cellular response detection in nasopharyngeal swab samples

PLOS ONE

Dear Dr. Mari,

Thank you for submitting your manuscript to PLOS ONE. After careful consideration, we feel that it has merit but does not fully meet PLOS ONE’s publication criteria as it currently stands. Therefore, we invite you to submit a revised version of the manuscript that addresses the points raised during the review process.

We look forward to receiving your revised manuscript.

Kind regards,

Tzong-Yueh Chen, Ph.D.

Academic Editor

PLOS ONE

Journal Requirements:

3.Thank you for stating the following in the Acknowledgments Section of your manuscript:

"Supported by funds from the “Centre National de la Recherche Scientifique” (CNRS), the “Université Côte d’Azur”, the French “French Defence Innovation Agency – Agence de l’Innovation de Défense “ (project “Safe and direct COV-2 qPCR Test”) and the Département des Alpes Maritimes (COVID-19 Health program). JF is supported by the Cancéropole PACA and CL is supported by Plan Cancer 2018 « ARN non-codants en cancérologie: du fondamental au translationnel » (number 18CN045). The Biomark equipment was funded by Canceropole PACA and France Génomique (Commissariat aux Grands Investissements: ANR-10-INBS-6 09–03, ANR-10-INBS-09–02)."

 " The funders had no role in study design, data collection and analysis, decision to publish, or preparation of the manuscript."

4.Thank you for stating the following in the Competing Interests section:

"No authors have competing interests"

We note that one or more of the authors are employed by a commercial company: Bayer SAS and LBM BIOESTEREL

Additional Editor Comments:

1. The data cannot support the conclusions. PLOS ONE is designed to communicate primary scientific research, and welcome submissions in any applied discipline that will contribute to the base of scientific knowledge. But the data of this manuscript cannot support the conclusions.

2. This manuscript needs to adhere the PLOS Data Policy. The authors need to make all methods, materials and data underlying the findings in their manuscript fully available.

3. The method must be of use to the community and must present a proven advantage over existing alternatives. If similar options already exist, the submitted manuscript must demonstrate that the new method is an improvement over existing options in some way. This requirement may be met by including a proof of principle experiment or analysis.

4. This manuscript has the statistical analysis problem.

5. The revised manuscript needs to address each of the comments of the reviewers.

Reviewers' comments:

Reviewer's Responses to Questions

**Comments to the Author**

1. Is the manuscript technically sound, and do the data support the conclusions?

Reviewer #1: Yes

Reviewer #2: Yes

2. Has the statistical analysis been performed appropriately and rigorously? 

Reviewer #1: Yes

Reviewer #2: No

3. Have the authors made all data underlying the findings in their manuscript fully available?

Reviewer #1: Yes

Reviewer #2: Yes

4. Is the manuscript presented in an intelligible fashion and written in standard English?

Reviewer #1: Yes

Reviewer #2: Yes

5. Review Comments to the Author

Reviewer #1: In this study, Dr. Bernard Mari and colleagues addressed a significant question regarding the two major obstacles of current COVID-19 test, the reagent shortage and tedious process of sample preparation. The authors delicately evaluated current approaches with a high-throughput platform in clinical samples. Overall, the data presented in the manuscript are of high quality with several important connections demonstrated, including accuracy and sensitivity. This manuscript should be great interest to a general audience, especially those countries are suffering from the massive COVID-19 test loading. In conclusion, I would suggest the editor directly accept this manuscript without any further revision.

Reviewer #2: The authors demonstrated a way to multiplex and bypass the RNA extraction for SARS-CoV-2 detection. The application of the IFC method is new and serve the purpose. However, the sample number of SARS-CoV-2 patients is low and damper the conclusive findings. I have several comments.

Specific comments

1. Increase the sample size of SARS-CoV-2 patients if possible.

2. Author should explain why to detect miRNA and the importance.

3. The label in figure 5 second row is quite confusing. Maybe change to VTM.

I don't understand the labeling in the figure 5 right part. DP? N? How you calculate sensitivity, by adding?

4. I won't say the 11 to 14 out of 17 is high. It is not acceptable for any applications.

5. The specificity was not evaluated in the lysis buffer and direct assay. Please revise.

6. The picture quality of the supplementary figures are very poor.

7. Please check all the labeling in text and figures and be consistent, such as SARS-CoV-2 and COVID-19.

8. Is there any conflict of interest?

6. PLOS authors have the option to publish the peer review history of their article (what does this mean?). If published, this will include your full peer review and any attached files.

Reviewer #1: No

Reviewer #2: No

---

## [Author Response · Author response to Decision Letter 0]

5 Mar 2021

Response to Reviewer's comments

- Reviewer #1: In this study, Dr. Bernard Mari and colleagues addressed a significant question regarding the two major obstacles of current COVID-19 test, the reagent shortage and tedious process of sample preparation. The authors delicately evaluated current approaches with a high-throughput platform in clinical samples. Overall, the data presented in the manuscript are of high quality with several important connections demonstrated, including accuracy and sensitivity. This manuscript should be great interest to a general audience, especially those countries are suffering from the massive COVID-19 test loading. In conclusion, I would suggest the editor directly accept this manuscript without any further revision.

We thank the reviewer for these very positive comments.

- Reviewer #2: The authors demonstrated a way to multiplex and bypass the RNA extraction for SARS-CoV-2 detection. The application of the IFC method is new and serve the purpose. However, the sample number of SARS-CoV-2 patients is low and damper the conclusive findings. I have several comments.

Specific comments

1. Increase the sample size of SARS-CoV-2 patients if possible. 

We agree with the reviewer that increasing the size of SARS-CoV-19 patient would improve the manuscript. However, this is not possible as our access to patient samples is limited, specifically for direct detection as we receive samples collected and transported in different VTM. As a result, this paper should, in our view, be regarded as a technical report and not necessarily as a fully validated clinical study.

2. Author should explain why to detect miRNA and the importance. 

It is unclear, at this stage, whether miRNA detection will be of any importance. However, we felt that being able to demonstrate that high-throughput miRNA detection is possible enriches the manuscript. We feel that this proof of principle adds value to the report and we have decided to keep this section in the revised manuscript.

3. The label in figure 5 second row is quite confusing. Maybe change to VTM. I don't understand the labeling in the figure 5 right part. DP? N? How you calculate sensitivity, by adding? 

We have modified the figure 5 (Figure 6 in the revised version of the manuscript) and we feel that this new version is more reader-friendly. The reviewer is right. We have calculated the sensitivity by adding the positive samples with the different probes. However, we omitted to define the acronym “DP” (which stands for double positive). This omission may have confused the reviewer. We have amended the legend of the figure accordingly.

4. I won't say the 11 to 14 out of 17 is high. It is not acceptable for any applications. 

We have removed the adjective “high”. The new sentence reads: “The R2 values dropped in a dramatic way for the two direct detergents based assays, with 11 to 14 positive samples detected out of 17” (lanes 469-470).

5. The specificity was not evaluated in the lysis buffer and direct assay. Please revise. 

We performed some assays on one negative sample and no-SARS-CoV2-specific signal was obtained in conditions of the Figure 5 (Figure 6 in the revised version). This is now mentioned in the legend of figure 6 in the revised manuscript.

6. The picture quality of the supplementary figures are very poor. 

 We have provided a new set of supplementary figures of a higher quality.

7. Please check all the labeling in text and figures and be consistent, such as SARS-CoV-2 and COVID-19. 

We have performed a few rounds of editing of the paper and are think that the text and figures are consistent.

8. Is there any conflict of interest? 

We do not report any conflict of interest (see point number 3 and 4 in the responses to the Editor’s comments).

---

## [Decision Letter · Decision Letter 1]

16 Mar 2021

PONE-D-20-36827R1

Versatile and flexible microfluidic qPCR test for high-throughput SARS-CoV-2 and cellular response detection in nasopharyngeal swab samples

PLOS ONE

Dear Dr. Mari,

Thank you for submitting your manuscript to PLOS ONE. After careful consideration, we feel that it has merit but does not fully meet PLOS ONE’s publication criteria as it currently stands. Therefore, we invite you to submit a revised version of the manuscript that addresses the points raised during the review process.

1. The author should have some sentences or a paragraph to describe the rationales, the findings, and the indications of miRNA. It is not a proper way to explain by merely adding more data.

2. Please check the SARS-CoV-2 and be consistent. The manuscript still see many SARS-CoV2 throughout the manuscript.

We look forward to receiving your revised manuscript.

Kind regards,

Tzong-Yueh Chen, Ph.D.

Academic Editor

PLOS ONE

Journal Requirements:

Reviewers' comments:

Reviewer's Responses to Questions

**Comments to the Author**

1. If the authors have adequately addressed your comments raised in a previous round of review and you feel that this manuscript is now acceptable for publication, you may indicate that here to bypass the “Comments to the Author” section, enter your conflict of interest statement in the “Confidential to Editor” section, and submit your "Accept" recommendation.

Reviewer #1: All comments have been addressed

Reviewer #2: (No Response)

2. Is the manuscript technically sound, and do the data support the conclusions?

Reviewer #1: Yes

Reviewer #2: Yes

3. Has the statistical analysis been performed appropriately and rigorously? 

Reviewer #1: Yes

Reviewer #2: Yes

4. Have the authors made all data underlying the findings in their manuscript fully available?

Reviewer #1: Yes

Reviewer #2: Yes

5. Is the manuscript presented in an intelligible fashion and written in standard English?

Reviewer #1: Yes

Reviewer #2: Yes

6. Review Comments to the Author

Reviewer #1: (No Response)

Reviewer #2: The authors have address most of my comments. There is a few comments still need to be addressed.

1. The author should have some sentences or a paragraph to describe the rationales, the findings, and the indications of miRNA. It is not a proper way to explain by merely adding more data.

2. Please check the SARS-CoV-2 and be consistent. I still see many SARS-CoV2 throughout the manuscript.

7. PLOS authors have the option to publish the peer review history of their article (what does this mean?). If published, this will include your full peer review and any attached files.

Reviewer #1: No

Reviewer #2: No

---

## [Author Response · Author response to Decision Letter 1]

19 Mar 2021

Responses to the reviewers 

Reviewer #2: The authors have address most of my comments. There is a few comments still need to be addressed.

1. The author should have some sentences or a paragraph to describe the rationales, the findings, and the indications of miRNA. It is not a proper way to explain by merely adding more data.

As recommended, we have added a paragraph in the discussion (lanes 523-532) describing the rationale, the findings and the indication of following miRNAs as biomarkers in nasopharyngeal swabs and cited 5 additional references to illustrate this point. 

2. Please check the SARS-CoV-2 and be consistent. I still see many SARS-CoV2 throughout the manuscript.

We have corrected these last typos.

---

## [Decision Letter · Decision Letter 2]

23 Mar 2021

Versatile and flexible microfluidic qPCR test for high-throughput SARS-CoV-2 and cellular response detection in nasopharyngeal swab samples

PONE-D-20-36827R2

Dear Dr. Mari,

We’re pleased to inform you that your manuscript has been judged scientifically suitable for publication and will be formally accepted for publication once it meets all outstanding technical requirements.

Kind regards,

Tzong-Yueh Chen, Ph.D.

Academic Editor

PLOS ONE

Additional Editor Comments (optional):

Reviewers' comments:

Reviewer's Responses to Questions

**Comments to the Author**

1. If the authors have adequately addressed your comments raised in a previous round of review and you feel that this manuscript is now acceptable for publication, you may indicate that here to bypass the “Comments to the Author” section, enter your conflict of interest statement in the “Confidential to Editor” section, and submit your "Accept" recommendation.

Reviewer #2: All comments have been addressed

2. Is the manuscript technically sound, and do the data support the conclusions?

Reviewer #2: Yes

3. Has the statistical analysis been performed appropriately and rigorously? 

Reviewer #2: Yes

4. Have the authors made all data underlying the findings in their manuscript fully available?

Reviewer #2: Yes

5. Is the manuscript presented in an intelligible fashion and written in standard English?

Reviewer #2: Yes

6. Review Comments to the Author

Reviewer #2: The authors have addressed all the issues.

7. PLOS authors have the option to publish the peer review history of their article (what does this mean?). If published, this will include your full peer review and any attached files.

Reviewer #2: No

---

## [Editor Report · Acceptance letter]

8 Apr 2021

PONE-D-20-36827R2 

Versatile and flexible microfluidic qPCR test for high-throughput SARS-CoV-2 and cellular response detection in nasopharyngeal swab samples 

Dear Dr. Mari:

I'm pleased to inform you that your manuscript has been deemed suitable for publication in PLOS ONE. Congratulations! Your manuscript is now with our production department. 

Kind regards, 

on behalf of

Prof. Tzong-Yueh Chen 

Academic Editor

PLOS ONE